# Low Phase-Noise, 2.4 and 5.8 GHz Dual-Band Frequency Synthesizer with Class-C VCO and Bias-Controlled Charge Pump for RF Wireless Charging System in 180 nm CMOS Process

**Jongwan Jo** [1,2], **David Kim** [1,2], **Arash Hejazi** [1,2], **YoungGun Pu** [1,2], **Yeonjae Jung** [1,2], **Hyungki Huh** [1,2], **Seokkee Kim** [1,2], **Joon-Mo Yoo** [1,2] **and Kang-Yoon Lee** [1,2,*]

1   Department of Electrical and Computer Engineering, Sungkyunkwan University, Suwon 16419, Korea; jw1114@skku.edu (J.J.); dkim9402@skku.edu (D.K.); arash@skku.edu (A.H.); hara1015@skku.edu (Y.P.); yjjung@skaichips.co.kr (Y.J.); gray@skaichips.co.kr (H.H.); skkim@skaichips.co.kr (S.K.); jmyoo@skaichips.co.kr (J.-M.Y.)
2   SKAICHips Co., Ltd., Suwon 16419, Korea
*   Correspondence: klee@skku.edu; Tel.: +82-31-299-4954

**Abstract:** This paper presents an integer-N phase-locked loop (PLL) for an RF wireless charging system. To improve the phase-noise characteristics under low power, a constant amplitude control class-C voltage-controlled oscillator (VCO) with a DC-DC converter, and a bias-controlled charge pump with a feedback loop are proposed. The frequency range of the VCO is 4.5–6.1 GHz, the target frequency of the proposed PLL is 2.4 and 5.8 GHz in the industry–science–medical band. It is designed with a same phase margin and bandwidth using one loop filter. The proposed PLL consumes less than 8 mW from a 1.8 V power supply with a settling time of fewer than 20 μs and an area of 1200 μm × 800 μm in the 180 nm CMOS process. For a carrier frequency offset of 1 MHz, the measured phase noise is −118.5 dBc/Hz at 2.4 GHz and −116.6 dBc/Hz at 5.8 GHz. Its FoM including the phase noise is −197 dB at 2.4 GHz and −202.8 GHz at 5.8 GHz, outperforming other PLLs designed in the 180 nm CMOS process.

**Keywords:** PLL; LC-VCO; class-C VCO; RF wireless charger; ISM band; charge pump

## 1. Introduction

Wireless chargers used in cell phones, wearables, and automobiles have grown significantly in popularity in recent years. With the rapid increase in wireless use of various mobile devices, charging users takes considerable time and effort. In addition, it is somewhat inconvenient to connect the power line to an electronic device or change the battery. Accordingly, a method of shortening the charging time of a mobile device and extending the battery life is attracting attention. In particular, research on wireless power transmission technology is being actively conducted, and accordingly, interest in a Wireless Power Transfer (WPT) system for supplying power to a battery is increasing.

There are two methods of WPT, a Near-field Power Transfer method of the WPC/A4WP method and a Far-field Power Transfer method of the RF method. Figure 1 shows the wireless charging technology [1–3]. Figure 1a shows the Near-field WPT, and Figure 1b shows Far-field WPT. The Near-field Power Transfer method uses a coil to transmit energy, and the standard method uses a frequency of 100–205 kHz, the magnetic induction method (WPC), and 6.765–6.795 MHz. There is a self-resonant method (A4WP) using a frequency band of the band. Far-field wireless charging is used to overcome the distance limit to the near-field frequency band by using frequencies above GHz, and there is currently no standard defined. In the case of Far-field WPT, there can be two advantages over Near-field WPT. One is to remove the distance limitation, and the other is to charge various devices

with one transmitter. Therefore, Far-field WPT is required to charge various devices, and the variation of the technology is expected to be large.

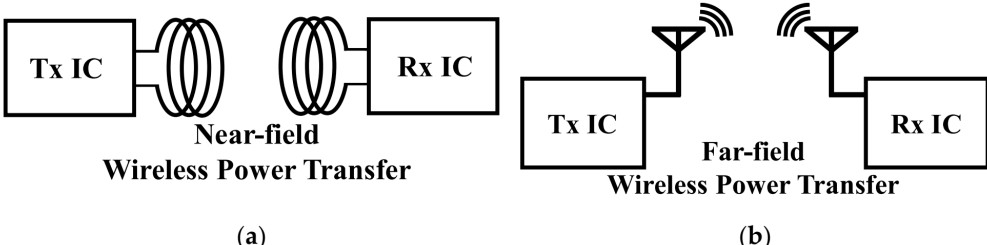

**Figure 1.** Wireless power transfer technique, (**a**) Near-field, (**b**) Far-field

Using RF wireless charging is an efficient way to remotely power nearby (a few meters away) smartphones, TVs, and Internet of Things (IoT) sensors. In the case of the RF wireless charging method, as shown in Figure 2, the beamforming method is used to charge the energy of the wireless power transmission method. When the beamforming method is used, a large amount of energy is transmitted. At this time, it is necessary to be able to select and transmit a different frequency according to the surrounding environment because it may affect the surrounding communication frequency [4]. In the case of a method that can transmit only the same frequency, there is a disadvantage in that it may cause frequency interference with other systems that use RF wireless charging power using different Industry-Science-Medical (ISM) bands and ISM bands. Supporting antenna and power amplifier research is in progress [5]. Conventional frequency Synthesizer is a method of using one or more VCOs operating in each band for research supporting a wide range of 2.4 GHz and 5.8 GHz, or a method of supporting two or more bands using two or more frequency Synthesizers. The advantage of the Dual Band frequency Synthesizer is that it supports the frequency according to the antenna arrangement, enables wireless power transmission at 2.4 GHz and 5.8 GHz, and reduces the area of the IC with a single frequency Synthesizer. Frequency synthesizers used in RF WPT systems must have a wide frequency range, small area, and low power consumption. In order to have a general phase noise with low power consumption in a frequency synthesizer, it is necessary to minimize interference with other communication frequencies through a technique for optimizing phase noise in the out-band.

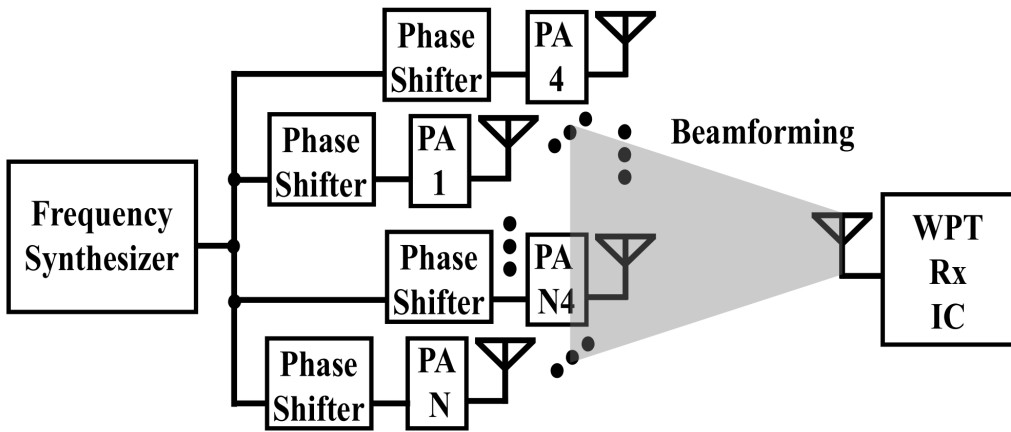

**Figure 2.** Conventional RF wireless power transfer technique.

The frequency synthesizer is a key block in the transceiver chain, and its performance outside the loop bandwidth is determined by the phase noise of the voltage-controlled oscillator (VCO). Low-voltage VCOs are an essential component of low-power wireless communications. The Class C VCO topology is superior to the Class B VCO topology due to



better phase noise performance at the same power consumption. However, startup robustness and amplitude stabilization are two important issues for Class C VCOs. Significant efforts have been made to overcome startup problems [6–8]. Two constant bias voltages were used in [6] for the starting and steady state. At steady state, a constant bias voltage degrades the robustness of the VCO due to PVT fluctuations. An analog feedback loop was proposed in [7,8]. Provides an adjustable bias voltage for the oscillator to ensure startup robustness and better performance in class C operation. However, these methods require additional current consumption, making them less energy efficient. In this study, a constant amplitude control class C-VCO that uses a DC-DC converter architecture to overcome the trade-off between startup robustness, amplitude stabilization, current consumption, and phase noise performance of a conventional class C VCO is: Utilizing the principle of constant amplitude control and on-chip switched-capacitor DC-DC converter, it proposes power-efficient normal operation.

Moreover, in phased-locked loop (PLL) structures with one loop filter, the current in the charge pump (CP) must be controlled to change the bandwidth and phase margin at different frequencies. Since the output current of CP affects the voltage control (VC) that is the input of the VCO, the output current noise of CP is also important to the overall phase noise of the PLL. To reduce the output noise, it is important to reduce the current mismatch of the CP. Furthermore, many CP structures used in PLLs use switches for current control. However, as these structures are stacked, slew rate can become an issue [9]. As the CP slew rate drops, the PLL's phase noise performance degrades because the desired current is not produced. Therefore, we propose a bias-controlled CP that utilizes a feedback structure and a high-speed op-amp to improve the slew rate and reduce the current mismatch.

## 2. Proposed PLL Architecture

In CP PLLs used in modern communication systems, the output phase of the VCO is divided by and compared with a reference phase in a phase-frequency detector (PFD). A CP current proportional to the phase noise is generated, low-pass filtered and applied to the control input of the VCO.

A block diagram of the proposed PLL is shown in Figure 3. It is a differential CP PLL architecture, which is commonly used in communications. The reference clock frequency is set to 50 MHz by using a crystal oscillator. This frequency and FDIV are compared by the PFD, which generates pulses proportional to the phase difference between the two input signals. If the reference frequency lags FDIV, UP pulses are generated, and the current is injected into the passive loop filter by the CP, increasing the control voltage and thus the frequency of VCO. If the reference frequency leads to FDIV, DN pulses are generated, and consequently, the frequency of the VCO decreases. To design a PLL of 5.8 and 4.8 GHz using one loop filter, the CP current and the VCO $K_{VCO}$ are set differently for each frequency. The output frequency range of the proposed VCO is 4.5–6.1 GHz to generate 4.8 and 5.8 GHz. To achieve an output frequency of 2.4 GHz, the VCO output frequency of 4.8 GHz is divided by two using a current-mode-logic (CML) divider at a high frequency [10]. For smooth operation of the prescaler, the VCO output is signaled by dividing it by 4 using the CML divider. In dividers in the gigahertz range, the signals driving the latch are more similar to sinusoidal waves than to square waves. Dependence of the output slope at rising and falling edges on the voltage gain of the differential pair is observed. That is, the voltage gain of the differential pair should be considered and controlled for an effective design. The prescaler operates at a lower frequency than the CML divider, and it is more difficult for the prescaler than for the CML to operate for all corners at high frequencies. In a pulse swallow frequency divider, the prescaler has two selectable division ratios, N and N + 1; it is combined with programmable counters to achieve a programmable division ratio [11,12].

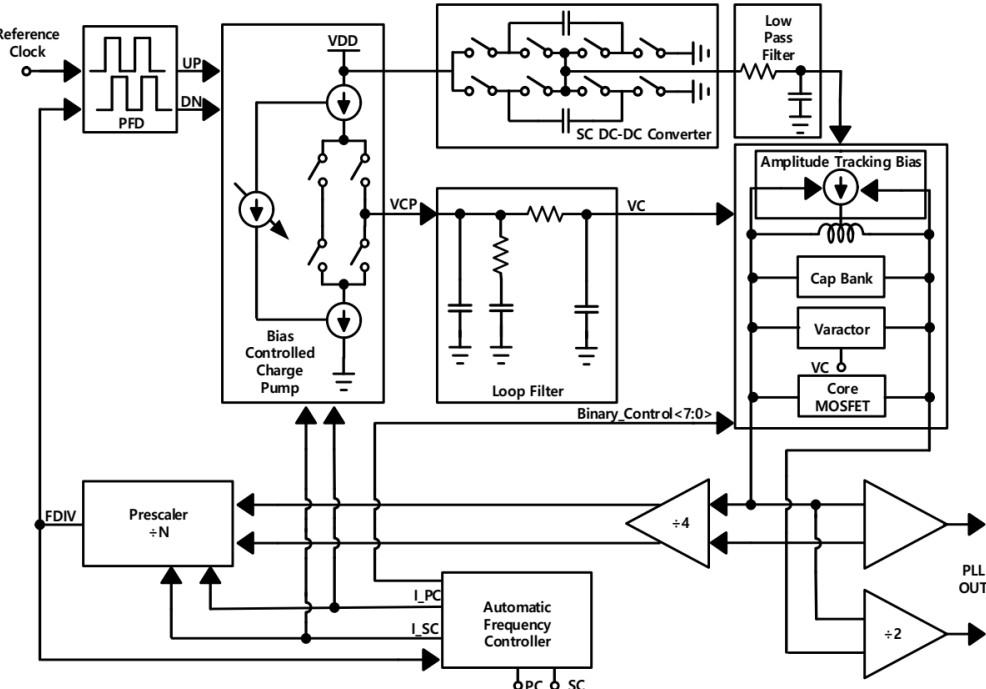

**Figure 3.** Proposed PLL block diagram.

### 3. Building Blocks

*3.1. Constant Amplitude Control Class-C VCO with DC-DC Converter*

RF WPT Phase Locked Loop VCO requires a loop for stable operation at various supply voltages and temperatures that may occur in WPT situations and a technique for minimizing power consumption to maximize energy efficiency. The proposed VCO is constructed through stable oscillation and optimized power supply to C Class VCO (Class-C VCO has higher phase noise than class-A and Class-B) [10]. To maintain the excellent DC–RF current conversion efficiency of transistors operating in class C and the maximum theoretical phase-noise figure of merit of the original class-C VCO, in some cases, a complementary LC tank oscillator is preferable to a single differential-pair oscillator [10]. The cross-coupled switching transistors, the AM-to-FM conversion noise of the varactors, and the thermal noise owing to losses inside the LC resonator are the main contributors to the phase-noise performance of a VCO. Varactor noise owing to AM-to-FM conversion can be lowered by decreasing $K_{VCO}$ and providing the frequency steps through the switch, thus compensating for capacitance. The proposed technique improves the frequency-tuning curve through the linearization of the varactor C (V) characteristic [13,14]. The designed varactor configuration consists of multiple varactor units, which are connected to the oscillator output at one end and to a resistor divider network at the other end. The C(V) characteristic of each varactor is voltage-shifted by the incremental voltage of the resistor divider network, and the net capacitance at the oscillator output has a more linear C(V) characteristic. The varactor provides continuous frequency tuning ranges of 60 and 100 MHz/V at center frequencies of 4.8 and 5.8 GHz, respectively. A higher VCO gain ($K_{VCO}$) requires a larger fixed capacitor inside the varactor and introduces large amplitude modulation into the frequency-modulated noise. Thus, the capacitance magnitude or the $K_{VCO}$ value of the varactor can be relaxed, and this capacitance can be compensated for by increasing the switch size in the coarse and intermediate tuning steps. The middle capacitor bank and varactor are located at the drain because the change in capacitance is smaller than at the gate. Two resistors are provided inside the switched capacitor to isolate the AC of the VCO node. To reduce the thermal noise of the switched capacitor internal resistance, size is optimized so that the noise contribution is less than 1%. The trade-off between the quality factor of a switched capacitor bank and the tuning range of the VCO is an unavoidable issue

with wide-tuning-range VCOs. The MOS switch introduces on-resistance into the signal path, lowering the quality factor of the tank. To lower the on-resistance of the switch, a large MOS transistor is required, considering the phase-noise reduction. Large MOS transistors introduce parasitic capacitance and limit the tuning range of the VCO. Therefore, small transistors are preferred for a wide tuning range. To reduce power consumption and phase noise, the unit capacitance in the capacitor banks is minimized, whereas the inductor size is as large as possible to achieve maximum swing with respect to the frequency range. The cap banks are binary weighted with linear frequency behavior and metal–insulator–metal (MIM) unit capacitors of 33 fF.

Figure 4 shows a constant amplitude control class-C VCO with DC-DC power. To simultaneously ensure startup robustness and amplitude stabilization, dual feedback loop solutions have been proposed [7]. Such a loop allows the VCO to achieve a highly robust startup as well as a constant amplitude for PVT variations or frequency tuning. For the negative feedback mechanism, $V_A$ and $V_B$ are the same as $V_{ref1}$ and $V_{ref2}$, respectively. Therefore, the oscillation amplitude $V_{osc}$ is fixed at a value slightly greater than $V_{ref1} - V_{ref2}$. When an amplitude increase $\Delta V_{osc}$ occurs, it is passed to the negative input of OTA, where it increases to $V_{bias2}$, and consequently, the current $I_{ref}$ decreases. This VCO also facilitates direct amplitude calibration methods, and therefore, it is highly reliable and stable in various applications. In addition, it offers adjustable bias voltages for the oscillators, ensuring startup robustness and better performance in class-C operation. When the amplitude detector (NM3, NM4) detects this large oscillation, the output $V_B$ follows a negative envelope that decays sharply to a level well below the $V_{ref2}$ level. When the differential input of the OTA becomes positive, charging starts, and $V_{bias2}$ rises gradually. When the amplitude control of the loop is activated, $V_{bias2}$ is changed, and then $I_{ref}$ is changed so that the oscillation amplitude $V_{osc}$ is automatically adjusted to the design value, as described above.

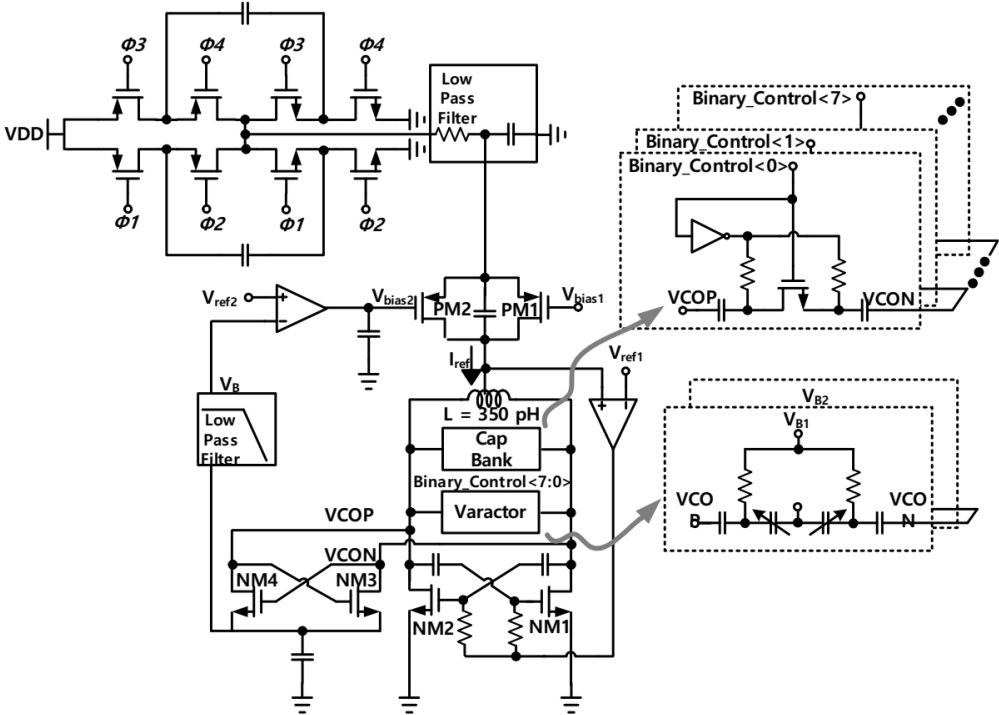

**Figure 4.** Schematic of the constant amplitude control class-C VCO with DC-DC converter.

In addition, this block contains a DC-DC converter to minimize power consumption. In this study, to reduce the power overhead of DC-DC conversion, a switched-capacitor converter is used to generate the VCO supply voltage [15]. It has high efficiency by design so that it can provide a low supply for the VCO without adding large spurs through

multiphase interleaving and careful switching frequency selection. By employing 4-phase interleaving, the switched-capacitor output ripple is reduced. To avoid this spur, the switched-capacitor DC-DC converter is designed to operate at the reference frequency of the PLL, hiding the DC-DC spur below the reference spur or using a method to minimize noise by connecting an LPF (Low Pass Filter) or LDO (Low Drop Out) [15]. In case of the pros and cons, if the DC-DC converter spur is hidden in the PLL reference clock spur, a high switching frequency of the DC-DC converter is required, which is not suitable for RF WPT. Therefore, it is necessary to minimize the spur of the DC-DC converter by using a low pass filter or LDO. Furthermore, the method of hiding the switching frequency spur of the DC-DC converter in the reference clock spur of the PLL has a disadvantage in that the effect of the spur is visible in the out-band. When an LDO is used, there is a disadvantage in that the power used for the additional amplifier is used, but it has the advantage of being able to design in a small area. In the RF WPT method, it was judged that it was not appropriate to use an LDO because it was necessary to minimize power for efficient power transmission. In the proposed Class-C VCO structure, the noise of the DC-DC converter is minimized through a low pass filter.

### 3.2. Bias-Controlled Charge Pump

The charge pump used in RF WPT adjusts the bandwidth by adjusting the loop filter value and the current of the Chare Pump to support both 2.4 GHz and 5.8 GHz frequencies. If you adjust the bandwidth through a charge pump other than the loop filter, the area used for the loop filter can be made smaller than when only the loop filter is used. Mismatch must be taken care of when changing the current of the charge pump [16,17]. In the case of the conventional charge pump used in the PLL, to reduce the mismatch, as shown in Figure 5a, the mismatch is minimized by using the same method as OTA. The method of minimizing the mismatch using the conventional OTA shown in Figure 5b requires more current consumption than the charge pump current in the OTA, and a large amount of current is not suitable for the RF WPT method.

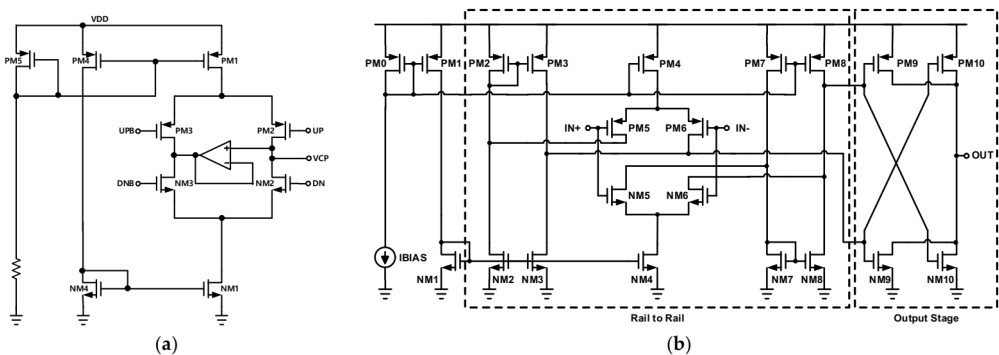

**Figure 5.** (**a**) Schematic of conventional Charge Pump based operational amplifier. (**b**) Rail to Rail OTA.

The circuit of the proposed CP is shown in Figure 6. The CP consists of a current-steering switch (NM2, NM3, PM2, and PM3) and a pumping current. It is a bias-controlled CP with an in-5-bit current bias. Its output current is controlled by changing the bias current; specifically, the output current is controlled from 0 to 1.6 mA in increments of 50 μA at each step to achieve the desired bandwidth and phase margin at each frequency.

The linearity of the CP current over the 32-step variation affects the accuracy of the loop bandwidth calibration. Thus, the size and layout of the MOSFETs used by the current mirror are carefully selected to minimize mismatches. Simulations demonstrate that the output current deviation from a perfect linear relation is maintained within 1%. As the loop filter terminal voltage approaches the rail potential (VDD or ground), the pumped up and down currents become unbalanced owing to channel length modulation effects. This causes the upper and lower pulse widths to be unequal in lockdown and induces a reference spur. To alleviate this, a tuned current feedback loop can be employed [18]. As shown in Figure 6,

the MOSFET size for PM4 and NM4 is the same size as PM3 and NM3, so it is possible to minimize the mismatch of the charge pump with half the current consumption compared to the charge pump method using the existing amplifier. Furthermore, through Current Control Based Frequency Detection, the same bandwidth can be obtained at 2.4 GHz and 5.8 GHz while minimizing the area of the loop filter.

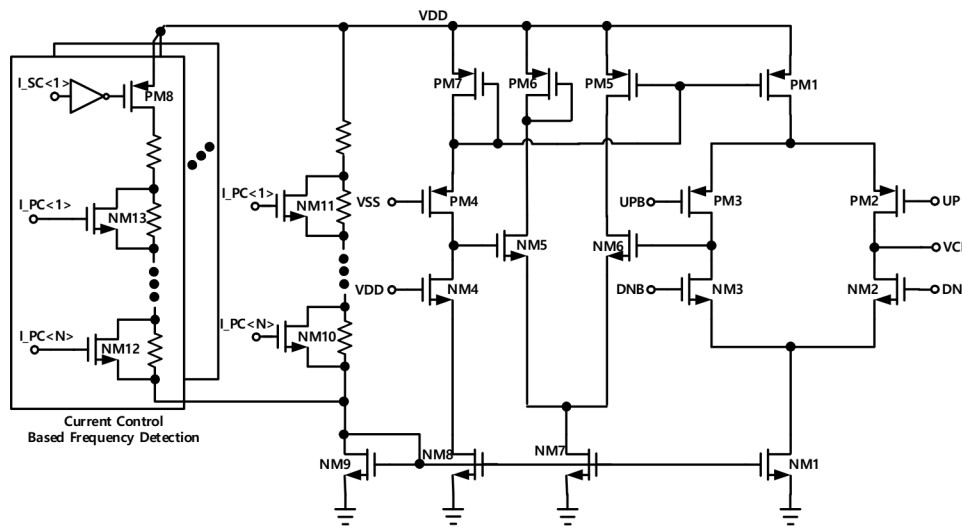

**Figure 6.** Schematic of the proposed bias-controlled CP.

## 4. Experimental Results

Figure 7 shows the layout of the proposed PLL, which has an area of 1200 μm × 800 μm. Figure 8a shows the measurement environment, and Figure 8b shows the evaluation board of the proposed PLL.

In the proposed VCO design, the VCO range for outputting the PLL output of 2.4 GHz and 5.8 GHz is from 4.8 GHz to 5.8 GHz, and after PLL Lock at 4.8 GHz, the frequency is divided by 2 to 2.4 GHz. As shown in Figure 9, the frequency range of the VCO ranges from 4.55 GHz to 6.6 GHz. The operating margin for stable operation of 4.8 GHz and 5.8 GHz, which are VCO frequency ranges, has sufficient range to accommodate the error range of the capacitor on the integrated circuit.

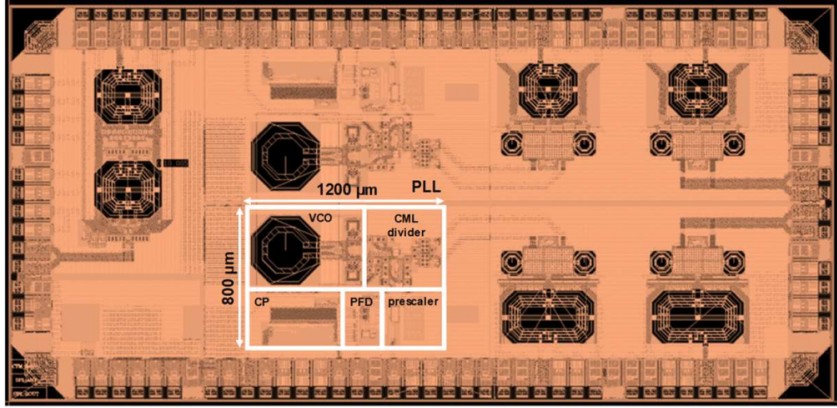

**Figure 7.** Layout of the proposed PLL.

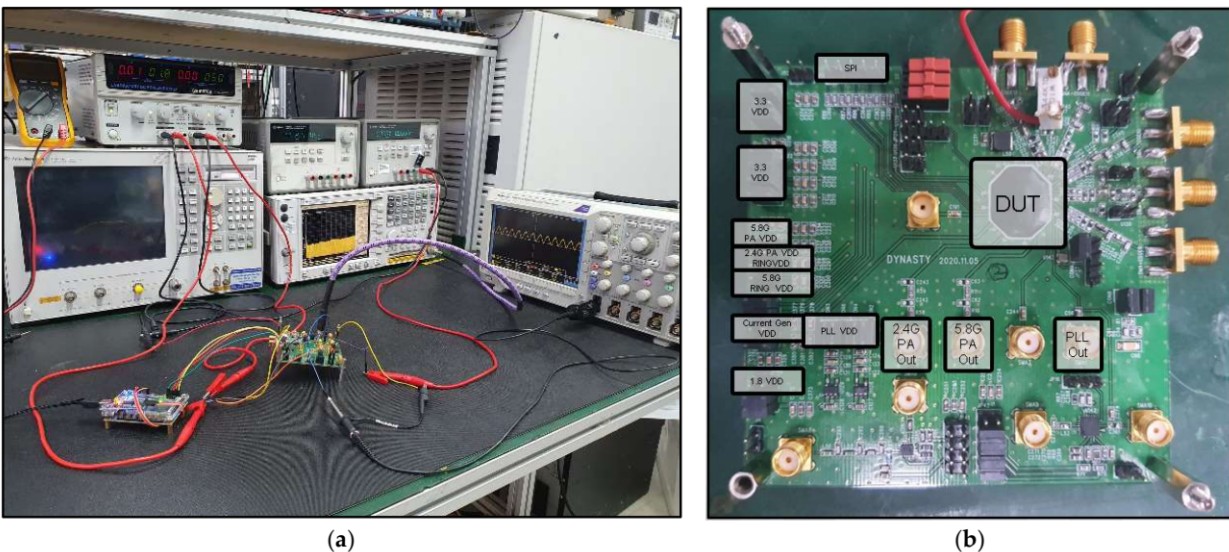

**Figure 8.** Measurement: (**a**) measurement environment; (**b**) evaluation board.

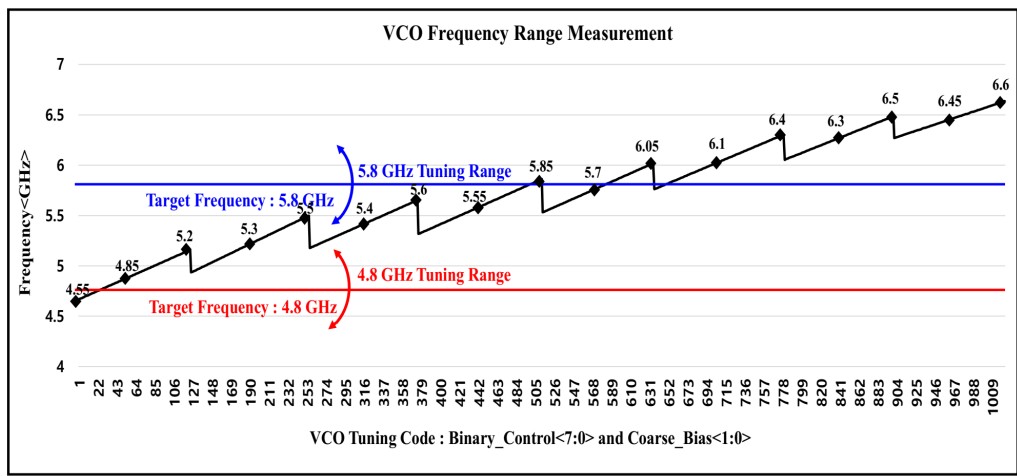

**Figure 9.** Frequency range of the VCO.

The difference in the proposed VCO phase noise can be confirmed in the measurement according to the ON/OFF of the switched DC-DC converter. When the Switched DC-DC Converter is on, the voltage that has passed through the LPF after the output of the Switched DC-DC Converter is applied to the VCO output. As shown in Figure 10a,b, at the VCO frequency of the 4.8 GHz band that brings the output of 2.4 GHz, the phase noise difference is about 3 dBc/Hz depending on the switched DC-DC converter ON/OFF. As shown in Figure 11a,b, in the 5.8 GHz VCO frequency band, phase noise improvement of about 5 to 10 dBc/Hz is obtained in the in-band and out-band.

Figure 12a,b show that for a carrier frequency offset of 1 MHz, the measured phase noise at a carrier frequency of 4.8 and 5.8 GHz is −112.5 and −116.6 dBc/Hz, respectively.

Through the low output current noise of the proposed CP and VCO, the measured phase noise confirms the improved performance for an offset frequency of 1 MHz. Table 1 shows a comparison of the phase-noise characteristics with those of other PLLs designed in the 180 nm CMOS process, indicating the superiority of the proposed PLL.

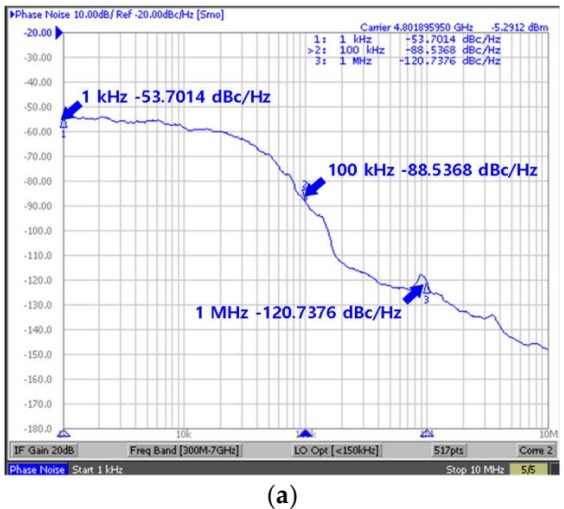
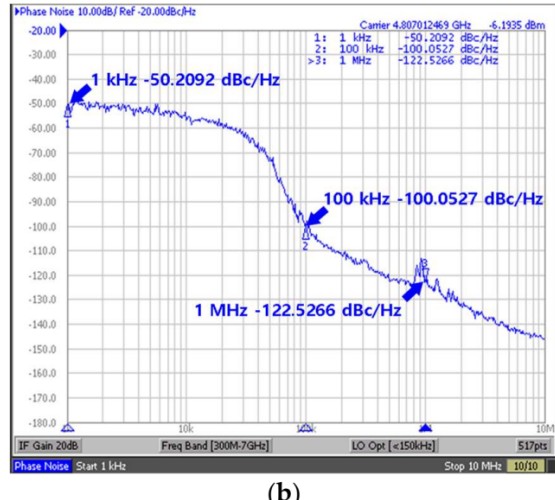

**Figure 10.** Output noise measurement result of the proposed VCO at 4.8 GHz (**a**) without switched dc-dc converter, and (**b**) with switched dc-dc converter.

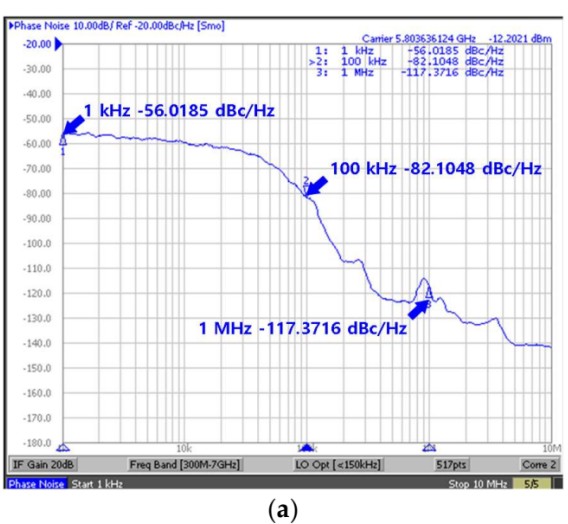
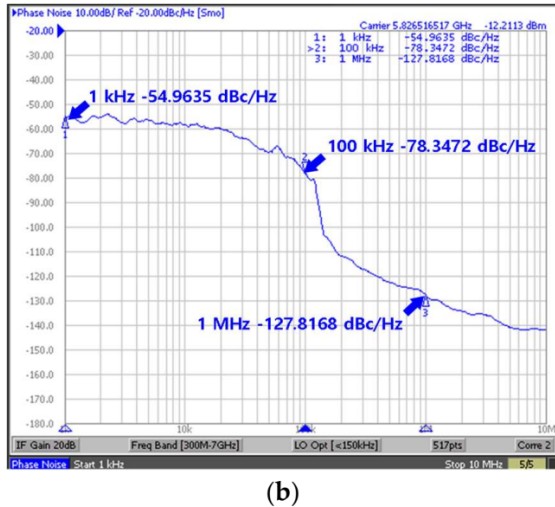

**Figure 11.** Output noise measurement result of the proposed VCO at 5.8 GHz (**a**) without switched dc-dc converter, and (**b**) with switched dc-dc converter.

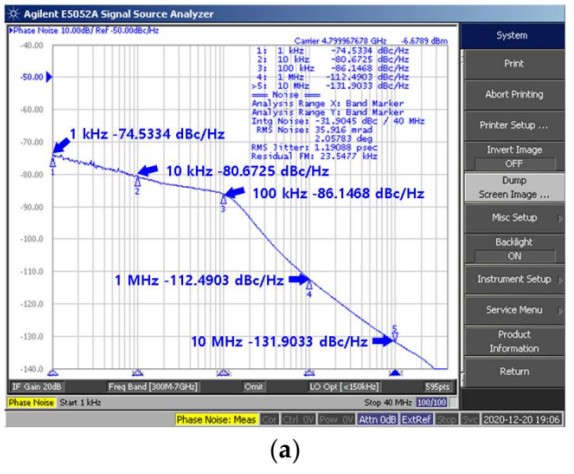
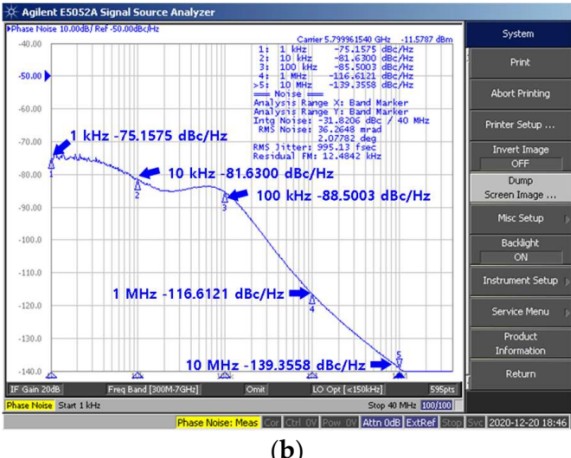

**Figure 12.** Measured phase noise of the proposed PLL at (**a**) 4.8 GHz, and (**b**) 5.8 GHz.

**Table 1.** Performance Comparison with Prior Works.

| Parameter | [19] | [20] | [21] | [22] | [23] | [23] | This Work |
|---|---|---|---|---|---|---|---|
| Technology | 130 nm CMOS | 65 nm CMOS | 65 nm CMOS | 130 nm CMOS | 40 nm CMOS | 40 nm CMOS | 180 nm CMOS |
| Supply Voltage (V) | 1.5 | 1.2 | 1.2 | 1.3 | 1 | 1/0.8 | 1.8 |
| Power Consumption (mW) | 12 | 3.9 | 15.5 | 21 | 1.19 | 1.7 | 8 |
| Bandwidth (kHz) | 100 | 1000 | 500 | 1500 | 1500 | 200 | 100 |
| Reference frequency (MHz) | 32.768 | 50 | 96 | 50 | 40 | 32 | 50 |
| Frequency range (GHz) | 5.7–6.0 | 5.0 | 6.0 | 2.12–2.4 | 1.7–2.7 | 1.7–2.7 | 4.5–6.1 |
| Phase noise (dBc/Hz) | −109 @1 MHz | −108 @1 MHz | −101 @1 MHz | −112 @1 MHz | −109 @1 MHz | −109 @1 MHz | −116.6/−118.5 @1 MHz |
| * FOM (dB) | −193.4 @5.8 GHz | −177.3 @5.8 GHz | −170.6 @5.8 GHz | −162.8 @2.4 GHz | −172.3 @2.4 GHz | −188.4 @2.4 GHz | −202.8 @5.8 GHz −197 @2.4 GHz |

* FOM = Phase Noise (dBc/Hz) − 20log ($\omega_0/\Delta\omega$) + 10log (Power(mW)).

## 5. Conclusions

The Proposed PLL is designed in the 180 nm CMOS process, with a power supply of 1.8 V and power consumption of 8 mW. In the case of the proposed PLL, one loop filter is used to cover two bands of 2.4 GHz and 5.8 GHz, and it is used in the RF Wireless Charging System. Its total area is 1200 µm × 800 µm. The frequency range of the VCO is 4.5–6.1 GHz. The settling time of the PLL is less than 20 µs, and its phase noise is −116.6 dBc/Hz at 5.8 GHz and −118.5 dBc/Hz at 2.4 GHz for a carrier frequency offset of 1 MHz. In the case of 2.4 GHz, 6 dBc/Hz degradation due to division by 2 of the 4.8 GHz PLL is reflected. Finally, the proposed PLL has an FOM of −202.8 dB and superior phase-noise characteristics compared with other PLLs designed using the 180 nm CMOS process.

**Author Contributions:** Conceptualization, J.J., D.K., A.H., Y.P., Y.J., H.H., S.K., J.-M.Y. and K.-Y.L.; formal analysis, J.J. and D.K.; project administration, A.H. and Y.P.; supervision, K.-Y.L.; writing—original draft, J.J.; writing—review and editing, Y.P., Y.J., H.H., S.K., J.-M.Y. and K.-Y.L. All authors have read and agreed to the published version of the manuscript.

**Funding:** This research received no external funding.

**Acknowledgments:** This work was supported by Institute of Information & communications Technology Planning & Evaluation (IITP) grant funded by the Korea government(MSIT) (No.2020-0-00261, Development of low power/low delay/self-power suppliable RF simultaneous information and power transfer system and stretchable electronic epineurium for wireless nerve bypass implementation).

**Conflicts of Interest:** The authors declare no conflict of interest.

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
