# Peer review of "Low Phase-Noise, 2.4 and 5.8 GHz Dual-Band Frequency Synthesizer with Class-C VCO and Bias-Controlled Charge Pump for RF Wireless Charging System in 180 nm CMOS Process"

_electronics, doi:10.3390/electronics11071118_

Round 1

Reviewer 1 Report

1. The manuscript is poorly written and extremely hard to follow. The manuscript should be thoroughly revised by a native English speaker and an expert who has published multiple technical papers. 

2. Also, based on the current manuscript, I do not find anything new. The authors should clarify what the major contribution of this work. 

I am willing to provide complete review comments if the authors could provide a full-revised manuscript with publishable quality. 

Author Response

First of all thanks to all our respected editor and reviewers for their comments and suggestions on our manuscript and giving us an opportunity to make it more clear and understandable.

Following revisions have been made based on the reviewer’s comments and suggestions.

Reviewer 2 Report

This paper presents a charge pump-based phase looked loop, whose interesting topology is well-described in its single blocks. The work is organized and written in a satisfactorily manner and extensive simulation and measurenent results are reported.

I have no particular comments for the authors. The presented work, in my opinion, ma well done and ccomplete of all the required information.

Only one m quastion is about dc-dc converter. It is known that switched capacitor circuits can work into two counterposed zones SSL and FSL (regulated charge pumps: a comparative study by means of verilog-ams). In what zone your CP works? There could differrnces in terms of noise be working into one limit rather than the other?

Author Response

First of all thanks to all our respected editor and reviewers for their comments and suggestions on our manuscript and giving us an opportunity to make it more clear and understandable.

It is SSL type and the noise issue only applies to one limitation.

Following revisions have been made based on the reviewer’s comments and suggestions.

Reviewer 3 Report

This paper proposes an integer-N phase-locked loop (PLL) for an RF wireless charging system. The novelty of the proposed solution is twofold: (1) a constant amplitude control class-C voltage-controlled oscillator (VCO) with DC–DC converter, and (2) a bias-controlled charge pump with a feedback loop.

A review of related literature would benefit the paper.

A mathematical model of the proposed transistor-level circuits would be advisable.

The conclusions are credible and support he hypotheses raised in the introduction. 

Author Response

(The authors gave the same response as above.)

Round 2

Reviewer 1 Report

None of my concerns have been addressed. The revised manuscript is identical to the original submission. The manuscript should be rejected. 

Author Response

We writed the part that compares our work with previous works to emphasize our advantage from line 26 to line 65 of page 1 as follows.

Thank you.

Best regards.

Round 3

Reviewer 1 Report

  1. In the first review, I asked the authors to thoroughly revise the manuscript in terms of English and technical contents, because it was poorly written and hard to follow. Although the authors have revised the introduction, however the remaining main body is still poorly written.
  2. I do not see any novelty from the manuscript. For example, the techniques used in the VCO (e.g. constant amplitude control) is not new. The charge-pump with a feedback is a well-known circuit. 
  3. In p5, the authors claims "To avoid this spur, the switched-capacitor DC–DC converter is designed to operate at the reference frequency of 
    the PLL, hiding the DC–DC spur below the reference spur." However, there is no evidence (i.e. reference spur measurement). The authors should provide the measured reference spur and show that the reference spur is good enough, in order to validate that the switching noise is hidden by the PLL reference spur while the PLL spur is not terrible. 
  4. The "Experimental results" section mostly enumerates the "simulation results". I accept Figure 7, but definitely Figure 8, 9 and 10 must be measured from the chip. Also, as mentioned in the comment #3, the measured reference spur should be included. 
  5. I believe this is a PLL paper, so the FOM metric used in this manuscript is not appropriate. It is to evaluate a VCO performance. 
  6. Based on the RMS jitter numbers from in Figure 11, the PLL FOM of the design is very poor. It is much worse than most of recent ring-PLLs, even though this work uses a LC. Given that the novelty of this work is weak, such a poor performance cannot be accepted. 
  7. DO NOT cheat in the comparison. AEUE is not a high reputation journal. The other IEEE publications were published more than 10 years before (2013, 2012, 2011, 2008!). 

Author Response

Thank you for your detailed review, and we have revised it to reflect all of them.
